# Postprandial Effects of Blueberry (*Vaccinium angustifolium*) Consumption on Glucose Metabolism, Gastrointestinal Hormone Response, and Perceived Appetite in Healthy Adults: A Randomized, Placebo-Controlled Crossover Trial

**DOI:** 10.3390/nu11010202

**Published:** 2019-01-19

**Authors:** Kim Stote, Adele Corkum, Marva Sweeney, Nicole Shakerley, Terri Kean, Katherine Gottschall-Pass

**Affiliations:** 1Division of Science, Mathematics and Technology, State University of New York, Empire State College, 113 West Avenue, Saratoga Springs, NY 12866, USA; 2Departments of Applied Human Sciences, University of Prince Edward Island, Charlottetown, PE C1A 4P3, Canada; acorkum@upei.ca (A.C.); kgottschall@upei.ca (K.G.-P.); 3Departments of Biology, University of Prince Edward Island, Charlottetown, PE C1A 4P3, Canada; msweeney@upei.ca; 4Department of Basic and Clinical Sciences, Albany College of Pharmacy and Health Sciences, Albany, NY 12208, USA; Nicole.Shakerley@acphs.edu; 5Faculty of Nursing, University of Prince Edward Island, Charlottetown, PE C1A 4P3, Canada; tkean@upei.ca

**Keywords:** blueberries, anthocyanins, glucose metabolism, gastrointestinal hormones, perceived appetite

## Abstract

The consumption of blueberries, as well as the phenolic compounds they contain, may alter metabolic processes related to type 2 diabetes. The study investigated the effects of adding 140 g of blueberries to a higher-carbohydrate breakfast meal on postprandial glucose metabolism, gastrointestinal hormone response, and perceived appetite. As part of a randomized crossover design study, 17 healthy adults consumed a standardized higher-carbohydrate breakfast along with 2 treatments: (1) 140 g (1 cup) of whole blueberries and (2) a placebo gel (matched for calories, sugars, and fiber of the whole blueberries). Each subject participated in two 2-h meal tests on separate visits ≥8 days apart. Venous blood samples and perceived appetite ratings using visual analog scales were obtained prior to and at 30, 60, 90, and 120 min after consuming the breakfast meals. Results show that glucose metabolism, several gastrointestinal hormones, glucagon-like peptide-1 (GLP-1), glucose-dependent insulinotropic peptide (GIP), peptide YY (PYY) concentrations and perceived appetite did not change significantly with blueberry consumption. However, pancreatic polypeptide (PP) concentrations were statistically significantly higher (*p* = 0.0367), and the concentrations were higher during 30, 60, 90, and 120 min after consumption of the blueberry breakfast meal than the placebo breakfast meal. Additional research is needed to determine whether blueberries and other flavonoid-rich foods reduce type 2 diabetes risk by modifying gastrointestinal hormones and perceived appetite.

## 1. Introduction

Globally, type 2 diabetes is a growing public health problem. Recent reports have indicated that close to 451 million people around the world are living with diabetes, primarily (90%) type 2 diabetes [1]. Type 2 diabetes is a serious metabolic disorder with a vast number of cases, which in turn has an impact on world health. At its simplest, it is characterized by hyperglycemia, due to hyposecretion of insulin from beta cells of the pancreas and/or reduced signaling at the insulin receptor. In reality there are more complicated causes involving genetics, obesity, over-eating and low levels of physical activity. Regardless, the failure to adequately control elevated blood glucose levels leads to serious complications such as nephropathy and retinopathy [2,3]. A meta-analysis of over 100 studies has shown that type 2 diabetes doubles the risk of cardiovascular disease incidence and increases overall mortality, making type 2 diabetes a leading cause of death [4].

Glucose control is the cornerstone of managing diabetes and preventing its dire complications. Such control can be achieved by (1) limiting the absorption of glucose from the gut, (2) regulating glucose production by the liver or muscles or enhancing the (3) secretion and (4) receptor sensitivity of insulin. Glucose absorption in to the blood is effectively reduced by eating smaller meals of low glycemic index. Because meal termination lowers total food intake, satiety is an important contributor to blood glucose regulation and body weight management, thereby reducing obesity [5]. Similarly, slower gastric emptying contributes to a more gradual glucose response after a meal. Certain foods have been shown to slow carbohydrate digestion and absorption, e.g., those high in fiber [6]. Flavonoid-rich diets (8 weeks treatment) may improve glucose metabolism in people with high cardiometabolic risk by inhibition of both intestinal glucose transporters (e.g., SGLTI, GLUT2) and digestive enzymes [7]. Insulin secretion is tightly regulated and can be induced by glucagon-like peptide 1 (GLP-1), a gastrointestinal hormone [8]. GLP-1 has multiple other anti-diabetic actions including slowing gastric emptying and suppressing the appetite (inducing satiety), thereby having anti-obesity effects [9]. Thus, the GLP-1 receptor is a target for diabetes drug development; anthocyanins may also modulate its activity and may help manage diabetes [10,11]. Finally, insulin resistance is strongly correlated with obesity and metabolic syndrome [12]. A cross-sectional study of 2000 women showed that higher anthocyanin intake was associated with less insulin resistance [13]. In addition to GLP-1, other gastrointestinal hormones such as glucose-dependent insulinotropic peptide (GIP), peptide YY (PYY) and pancreatic polypeptide (PP) may have an impact on glycemic control by inducing satiety and meal termination [5]; insulin also induces satiety [14]. To our knowledge, no human clinical trials have evaluated the effects of blueberries on gastrointestinal hormones. Blueberries are an abundant dietary source of flavonoids, specifically anthocyanins, which make up >50% of total phenolic compounds in blueberries, followed by hydroxycinnamic acid derivatives, flavonols, and flavanols [15].

The purpose of the study was to evaluate postprandial glucose metabolism, gastrointestinal hormone response, and perceived appetite following a higher-carbohydrate breakfast meal with or without whole blueberries (140 g), using a randomized crossover design with a placebo control gel, in healthy adults. We hypothesize that study participants consuming blueberries with a higher-carbohydrate meal will have beneficial effects on the dependent variables.

## 2. Materials and Methods

### 2.1. Participants

Healthy men and women aged 22 to 65 years were recruited by advertisement from the greater Charlottetown, Prince Edward Island, Canada area from April 2017 to May 2017. Inclusion in the study was based on being free of serious illness; a BMI ≤30 kg/m^2^; and blood pressure <150/100 mm Hg. Exclusion criteria were participants who: reported tobacco use; have given birth during the previous 12 months; are pregnant or planning to become pregnant during the study, lactating, or initiating or changing a hormone replacement therapy regimen within 3 months of the start of the study; have a history or presence of kidney disease, liver disease, gout, certain cancers, thyroid disease, gastrointestinal disease, other metabolic diseases, or malabsorption syndromes; have type 1 and 2 diabetes requiring the use of oral anti-diabetic agents or insulin; have a history of eating disorders; have a history of drug and alcohol abuse; are routinely participating in a heavy exercise program or initiating an exercise program during the study; having lost 10% of body weight within the past 12 months or planning to initiate weight loss; or have a known (self-reported) allergy or adverse reaction to blueberry products. Healthy adults were selected as study participants as the absorption, distribution, metabolism and excretion of flavonoids in blueberries may be affected by disease states such as type 2 diabetes. A nurse practitioner approved study entry based on the participants’ self-reported medical history and review of current blood pressure and weight status. Participants gave their informed consent to participate, and the University of Prince Edward Island, Research Ethics Board approved the experimental protocol. The participants received $200 for their successful participation in the study. The trial was registered at clinicaltrials.gov (NCT03192605). This study was approved by the University of Prince Edward Island, Research Ethics Board and written informed consent was obtained from all participants.

### 2.2. Study Design

The study was a randomized, 2-period crossover design in which study participants completed 2 postprandial breakfast meal tests that consisted of single-day visits which lasted approximately 3 h each. Each postprandial breakfast meal test was separated by ≥8 days. The randomization plan was generated with the use of the Second Generator Plan from randomization.com prior to the start of the study [16]. Participants were assigned to the randomization plan in order of recruitment. The study investigators generated the randomization plan, then enrolled and assigned participants to the interventions. The study was conducted at the University of Prince Edward Island, Human Nutrition Research Center, Charlottetown, Prince Edward Island, Canada. Participants were randomly assigned to consume a standard higher-carbohydrate breakfast meal along with 2 treatments: (1) 140 g (1 cup) whole blueberries (frozen, removed from freezer within 1 h of consumption) and (2) a placebo gel (color/flavor/energy/fiber-matched). The participants were not blinded to the treatments as it is difficult to blind food treatments. Participants who received blueberries for the first treatment were given the placebo for the second treatment and vice versa.

### 2.3. Study Treatments

The blueberries used in this study were the Canadian wild (lowbush) blueberry variety, *Vaccinium angustifolium*. Frozen blueberries were purchased in bulk as 1 lot from Atlantic Superstore (Charlottetown, Prince Edward Island, Canada). The blueberries were stored at −18 °C until use. Samples of the blueberries were collected at the end of the study and stored at −18 °C until nutrient and polyphenol analysis. Proximate and sugar profile were determined by the Research and Productivity Council Science & Engineering, Fredericton, New Brunswick, Canada. Anthocyanins were determined by the spectrophotometric pH differential and Folin–Ciocalteu method, respectively, at the University of Prince Edward Island Human Nutrition Research Center [17]. The placebo gel was developed at the University of Prince Edward Island Human Nutrition Research Center to match features similar to the blueberries but without polyphenols. The placebo gel contained water, agar, colorants (red), artificial blueberry flavor, inulin, glucose, and fructose. Nutrient composition of the treatments is shown in Table 1. 

The standard higher-carbohydrate breakfast meal, without blueberries or placebo gel, contained 290 kilocalories with 76% of energy from carbohydrate (55 g carbohydrate, 4.2 g protein, 6 g fat, and 1 g fiber). The breakfast meal consisted of 70 g of waffles and 30 ml maple syrup. 

### 2.4. Study Day Visit Procedures

Participants were instructed by a registered dietitian/nutritionist on eliminating consumption of anthocyanin-containing foods 7 days prior to starting the study and throughout the study duration (including the washout period). To achieve this, participants were asked to refrain from consuming berries and grapes and juices that contained them, as well as wine. Dietary intake data were collected and analyzed by nine, 24-h dietary recalls throughout the study duration, including the washout period, using the Automated Self-Administered 24-h Recall (ASA-24)-Canada-2016 system [18]. The day prior to the study day, participants were instructed to consume a similar evening meal (approximately 15% protein, 55% carbohydrate, 30% fat); increase water consumption (1500–2000 mL); and consume all their food before 8 p.m. On each study day, participants arrived at the University of Prince Edward Island, Human Nutrition Research Center, in the morning, after a 12-h overnight fast. After standard admission procedures (blood pressure collection, body weight and composition using bioelectrical impedance analysis, and a general health assessment) were completed, a fasting/baseline blood sample was collected. Subsequently, participants consumed the breakfast meal with either the blueberries or placebo gel within 20 min according to their randomly assigned sequence. The participants were allowed to drink approximately 200 mL of water with their breakfast meal. After consuming the breakfast meal, a registered nurse placed an intravenous catheter in the antecubital vein in the participant’s non-dominant arm and blood samples were collected at time points 30, 60, 90 and 120 min to assess changes in glucose metabolism (glucose and insulin) and gastrointestinal hormones (GLP-1, GIP, PYY, and PP) after the meal. In addition, perceived appetite ratings were assessed using a visual analog scale (VAS) ratings of hunger, fullness, desire to eat, and prospective food consumption using a 100-mm scale ranging from 0 (“not at all”) to 100 (“extremely”) [19]. Participants were asked to complete the VAS questionnaire before and every 30, 60, 90 and 120 min after consumption of the breakfast meal. The participants were allowed to sit, read, and talk quietly during the 2-h intervention. At the end of the 2-h intervention, the intravenous catheter was removed, and participants were evaluated for wellbeing prior to leaving the research facility.

Participants completed a daily questionnaire regarding their general health; any consumption of prescription and over-the-counter medications; factors related to dietary compliance; and exercise performed. Participants were encouraged to maintain their normal exercise routine throughout the study. No participants were on hormone replacement therapy or oral contraceptives. In addition, participants were asked to abstain from using any non-prescription drugs, vitamin, and dietary supplements for at least 7 days prior to the start of the study and throughout the study duration.

### 2.5. Biological Sample Collection and Analysis

Blood samples were collected by NaFl vacutainer tubes, centrifuged for 10 min at 1300 *g* at 4 °C, and by serum separator vacutainer tubes, allowed to clot at room temperature for 30 min before centrifugation. Additional blood samples were used to prepare 0.8–1.0 mL aliquots of plasma from BD P800 (BD, Franklin Lakes, NJ, USA) vacutainer tubes that were stored at −80 °C until analysis. BD P800 vacutainer tubes containing protease, esterase and DPP-IV inhibitors were used in the collection of blood for analysis of gastrointestinal hormones. Plasma glucose concentrations were measured using an Abbott Aeroset Chemistry Analyzer and serum insulin concentrations were measured by immunoassay with chemiluminescent detection on a Siemans Immulite 2000 XPi Immunoassay System. Analyses of glucose and insulin concentrations were performed at Queen Elizabeth Hospital, Charlottetown, Prince Edward Island, Canada. GLP-1 (active), GIP, PYY (total), and PP concentrations were determined using the MILLIPLEX MAP Human Gut Hormone Panel (EMD Millipore, Merck KGaA, Darmstadt, Germany). Analyses of gastrointestinal hormone concentrations were performed at Albany College of Pharmacy and Health Sciences, Albany, NY, USA. All analytes were measured in duplicate. The mean of the analytes was used for statistical analyses.

### 2.6. Statistical Analysis

All statistical analyses were performed using SAS, version 9.4 (SAS Institute, Cary, NC, USA). Data were analyzed by analysis of covariance (ANCOVA) appropriate for a 2-period, 2-treatment crossover study with baseline values prior to each period and repeated measures within periods. The full statistical model included treatment sequence, period (first or second period), treatment group, time, the treatment group by time interaction, and covariates, BMI, gender, weight at enrollment, height, age, and percent body fat, as fixed effects. Subject nested in sequence was included in the model as a random effect. Where the covariate was not statistically significant, it was dropped from the model. The relationship between outcome and treatment sequence, period, and treatment group was re-evaluated, including any covariate where *p* < 0.05. If the treatment by time interaction was significant (*p* < 0.05), within time treatment effects were evaluated. If this interaction was not significant, the main effect of treatment was assessed. Data were tested for normality with the Shapiro-Wilk statistic and visual inspection of residual plots. The effect of treatment was assessed at *p* < 0.05. The area under the curves of glucose, insulin, and gastrointestinal hormone time courses were calculated with the use of the trapezoidal method. Data are presented as least squares means ± SEMs. 

Sample size for this study was based on earlier studies of postprandial responses that reported a significant reduction in glucose with a similar human study intervention [20,21,22,23,24]. Therefore, sample size was determined by designing the trial to have >80% power and an α of 0.05 to detect a significant difference in glucose with a minimum of 16 participants [25,26].

## 3. Results

### 3.1. Baseline Subject Characteristics

Of the 64 persons who attended the study information meetings, 64 signed an informed consent form, and 64 completed the screening process. Forty-seven individuals were excluded; 41 did not meet the inclusion criteria and 6 declined to participate. Ultimately, 17 participants were randomly assigned to the treatments (Figure 1). The baseline physical characteristics of the 17 participants included in the final data are presented in Table 2. 

Mean daily energy intake was 1972 ± 110 kcal and 1990 ± 110 kcal for the blueberry and placebo groups, respectively, and did not differ significantly among the 2 treatment groups. There were no significant differences in treatment groups on changes in carbohydrate, protein, fat, cholesterol or fiber intake, as determined by the ASA24-Canada-2016 24-h recall method, during the course of the intervention. In addition, review of 24-h dietary recalls showed an avoidance of high-anthocyanin foods throughout the study period. Anthocyanins and other nutrients in the treatments were found to be stable at the end of the study when compared to analyses at the beginning of the study. Body weight did not change significantly throughout, as shown by the study pre- and post-treatment means 64.3 ± 13.1 kg compared to 64.6 ± 13.1 kg, respectively (*p* = 0.8561).

### 3.2. Glucose Metabolism, Gastrointestinal Hormone Response, and Perceived Appetite

Plasma glucose concentrations were higher than baseline at 30 min, then dropped below the fasting concentration after the consumption of both blueberry and placebo breakfast meals (*p* < 0.0001; Figure 2). Serum insulin concentrations were higher than baseline at all time points after the consumption of both blueberry and placebo breakfast meals (*p* < 0.0001; Figure 2). The glucose and insulin concentrations peaked at 30 min post breakfast meals and were lower thereafter. There were no statistically significant differences between the 2 treatments with breakfast meals for glucose (*p* = 0.2956; Figure 2) concentrations; however, insulin concentrations at 30 and 60 min were lower in the blueberry treatment group (−9% and −16%, respectively), although this did not reach statistical significance (*p* = 0.3994; Figure 2). The difference between treatments for glucose iAUC was not significantly different (575 vs. 595 mmol min/L, blueberry vs. placebo, respectively, *p* = 0.2974). After the blueberry breakfast meal, the insulin iAUC was −9% less than the placebo breakfast meal, though the differences did not reach statistical significance (14738 vs. 16196 µU min/L, blueberry vs. placebo, respectively, *p* = 0.3965). BMI, gender, weight at enrollment, height, age, and percent body fat were not significant covariates for glucose and insulin results.

Plasma GLP-1 and GIP concentrations were higher from baseline at 30 min, then were lower after the consumption of both blueberry and placebo breakfast meals (*p* < 0.0001; Figure 3). There were no statistically significant differences between the blueberry and placebo breakfast meals for GLP-1 (*p* = 0.5572; Figure 3) and GIP (*p* = 0.2272; Figure 3) though GIP concentrations at 60 min were −22% lower in the blueberry treatment group. Plasma PYY concentrations were higher from baseline at 60 min, then were lower thereafter (*p* < 0.0001; Figure 3). There were no statistically significant differences between the blueberry and placebo breakfast meals for PYY (*p* = 0.5998; Figure 3). Plasma PP concentrations were higher from baseline at 30 min, then were lower after the consumption of both blueberry and placebo breakfast meals (*p* < 0.0001; Figure 3). The overall difference in plasma PP concentrations between the treatments were statistically significant (*p* = 0.0367; Figure 3) and the PP concentrations were higher during 30 min (15%), 60 min (11%), 90 min (21%) and 120 min (10%) after consumption of the blueberry breakfast meal than the placebo breakfast meal. BMI, gender, weight at enrollment, height, age, and percent body fat were not significant covariates for the gastrointestinal hormones results.

Perceived appetite, using visual analog scale (VAS) ratings, were similar in response to the two treatments with breakfast meals for hunger (*p* = 0.9214), fullness (*p* = 0.5305), desire to eat (*p* = 0.9891), and prospective food consumption (*p* = 0.3701). There were statistically significant effects of time for hunger (*p* = 0.0006), fullness (*p* < 0.0001), desire to eat (*p* = 0.0002), and prospective food consumption (*p* = 0.0002) (Figure 4). Participants’ body fat was a statistically significant covariate where scores were higher with increased body fat for hunger (*p* = 0.0412), fullness (*p* = 0.0384), desire to eat (*p* = 0.0105), and prospective food consumption (*p* = 0.0339). BMI, gender, weight at enrollment, height, and age were not significant covariates for the perceived appetite results. 

## 4. Discussion

This randomized, placebo-controlled, crossover design human intervention study is among the first to assess the effects of consuming approximately 140 g (1 cup) of whole blueberries with a higher-carbohydrate breakfast meal on postprandial glucose metabolism, gastrointestinal hormone response, and perceived appetite in healthy adults. Results showed that glucose metabolism, several gastrointestinal hormones (GLP-1, GIP, and PYY) and perceived appetite did not change significantly with blueberry consumption. However, plasma PP concentrations were significantly higher at 30, 60, 90, and 120 min after consumption of the blueberry breakfast meal than the placebo breakfast meal.

No significant differences in glucose metabolism, despite a trend for lowered early insulin release at 30 and 60 min, were found between the blueberry and placebo breakfast meals. There is positive preclinical evidence that blueberries may improve glucose metabolism, even though our study produced negative results. Blueberry extracts reduced plasma glucose 6 h after ingestion in mice [27]. Blueberry anthocyanins stabilized in soybean flour also reduced glucose intolerance in mice and decreased glucose production in rat hepatocytes [28]. Further, blueberry extracts were found to induce beta-cell proliferation in vitro [29]. Few human clinical trials have evaluated the postprandial effects of blueberries on glucose and insulin response in healthy adults. In agreement with our study, a previous human study with blueberries along with a higher-carbohydrate meal (50 g carbohydrate, 13 g protein, 7 g fat) did not alter glucose response in healthy adults. Although the amount of blueberries with the specific number of anthocyanins required to improve glucose metabolism is unknown, this previous study used 50 g of blueberries compared to the 140 g of blueberries used in our study [20]. Per our study participants’ comments, consuming more than 140 g of whole blueberries with a meal may not be feasible for typical consumption. A recent study showed that consumption of anthocyanin-rich blueberry powder beverages (containing 310 mg or 724 mg anthocyanins) versus an equivalent sugar dose beverage significantly extended the postprandial glucose response in healthy young adults; though the placebo was not matched for fiber which could have resulted in the significant effects observed [30]. Several acute human clinical trials with 150 g of mixed berries (strawberries, bilberries, lingonberries, and chokeberries), consumed as a puree or juice, reported significant reductions of postprandial glucose and insulin in healthy adults [21,22,23,24]. Longer human clinical trials of blueberry consumption on glucose metabolism have produced mixed findings. Consumption of 25–50 g of freeze-dried blueberry powder (equivalent to approximately 140280 g fresh) for 6–8 weeks had no significant effect on glucose metabolism and surrogate markers of insulin sensitivity in adults with metabolic syndrome and risk factors for cardiovascular disease [31,32,33]. However, daily consumption of 45 g of freeze-dried blueberry powder for 6 weeks improved insulin resistance in men and women with insulin resistance [34]. More research is required to determine the long-term effects of blueberry consumption on glucose metabolism with mixed meals that include all macronutrients.

Gastrointestinal hormones may contribute to hunger, satiety, and metabolic responses, such as glucose metabolism. The main target of these hormones are the neurons in the arcuate nucleus region of the hypothalamus. There are 2 distinct populations of neurons with opposing actions releasing orexigenic (appetite-stimulating) peptides and anorexigenic (appetite-inhibiting) peptides. The released peptides act as neurotransmitters to produce the feeling of either hunger or satiety [5,35,36]. We found that the blueberry breakfast meal significantly increased postprandial PP concentrations more than the placebo breakfast meal, though the gastrointestinal hormones GLP-1, GIP, and PYY, did not significantly change. PP is secreted from the pancreatic islet F cells in response to food intake, and as a result there is an increase in PP for approximately 30 min after meal consumption [5]; we found this result in our study. PP is frequently included in the list of gastrointestinal hormones due to close connection between the pancreas and gastrointestinal tract. PP is considered an anorexigenic hormone and may decrease gastric emptying along with lessening pancreatic exocrine secretions. PP is hypothesized to play a role in satiation and satiety [5]. Peripheral administration of PP in rodents reduces food intake and suppresses hunger in the short term. Similarly, in normal weight adults, peripheral administration of PP decreases food intake by approximately 25% [37]. To our knowledge, no human clinical trials have evaluated the effects of blueberries on gastrointestinal hormones. However, some human clinical trials have shown significant effects of berries on the incretins GLP-1 and GIP [23,38,39]. GLP-1 is secreted by the ileum L cells and GIP is secreted by the duodenum K cells in the gastrointestinal tract; GLP-1 and GIP stimulate glucose-dependent insulin secretion and insulin biosynthesis, which controls postprandial glucose disposal [8,40]. Blackcurrant extract (equivalent to 100 g fresh blackcurrants, 599 mg anthocyanins) with a high-carbohydrate meal decreased postprandial GLP-1 and GIP, and attenuated glucose and insulin concentrations in healthy adults [39]. Similarly, we showed the blueberry breakfast meal decreased GIP concentrations. Though it did not reach statistical significance, this finding could be due to a lower dose of anthocyanins used in our study. The reduction of GLP-1 and GIP may be a key mechanism for the inhibition of early-phase insulin secretion [8]. Consumption of 150 g of mixed berries (strawberries, bilberries, lingonberries, and chokeberries), as a puree or juice, had a modest effect on GLP-1 concentrations in healthy adults [23]. Further acute, short-term, and long-term clinical trials are warranted to determine how blueberries and other flavonoids affect gastrointestinal hormones in humans. 

Perceived appetite, using VAS ratings of hunger, fullness, desire to eat, and prospective food consumption were similar in response to the 2 treatments with breakfast meals. Interestingly, participants’ body fat was a significant covariate where hunger, fullness, desire to eat and prospective food consumption scores were higher with increased body fat. It is important to note that our study controlled for dietary fiber in both treatments. This may explain the observed results of our study as dietary fiber promotes satiety and alters the secretion of some gastrointestinal hormones; even small amounts of fiber, such as 2.5 g, have been shown to affect perceived appetite [41]. Few human clinical trials have evaluated the effects of blueberries on perceived appetite using VAS. Though, similar to our findings, a 160-g mixed berry snack composed of strawberries, raspberries, blackberries, and blueberries compared to an isoenergetic confectionary snack had no effects on perceived appetite with VAS in healthy young women. However, the mixed berry snack decreased subsequent energy intake during the evening meal [42]. Unfortunately, our study did not measure subsequent food intake throughout the day; further research may be justified especially in those with increased body fat.

The strengths and limitations of our study design warrant consideration. To our knowledge, this is the first human study that evaluated the acute effects of whole blueberries compared to a placebo matched for kilocalories, sugars and fiber with a randomized, crossover design; this is one of the most powerful designs for evaluating the efficacy of dietary treatments [43]. There were, however, several limitations to the study. The small sample size was limiting, and the study may not have been adequately powered to detect existing differences in some biomarkers and perceived appetite ratings. We considered that this study with blueberries would produce possible differences in the study outcomes, due to results of previous acute human berry clinical trials evaluating glucose metabolism [21,22,23,24,30,38,39]. With that said, longer human clinical trials with a larger sample size are necessary. 

## 5. Conclusions

Consumption of 140 g of blueberries with a higher carbohydrate breakfast meal positively affected the plasma PP concentrations which may have an impact on reducing food intake by inducing satiety and meal termination. However, glucose metabolism, the gastrointestinal hormones GLP-1, GIP, PYY and perceived appetite were not significantly altered by the blueberry treatment. Additional research, such as mid- to long-term human clinical trials, are needed to determine whether blueberries and other flavonoid-rich foods reduce type 2 diabetes risk by modifying gastrointestinal hormones and perceived appetite. 

## Figures and Tables

**Figure 1 nutrients-11-00202-f001:**
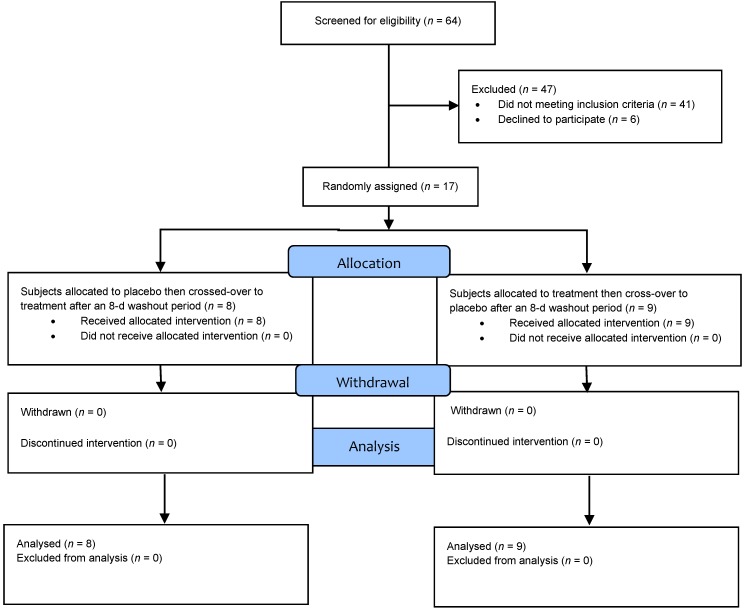
CONSORT diagram for study trial. CONSORT, Consolidated Standards of Reporting Trials.

**Figure 2 nutrients-11-00202-f002:**
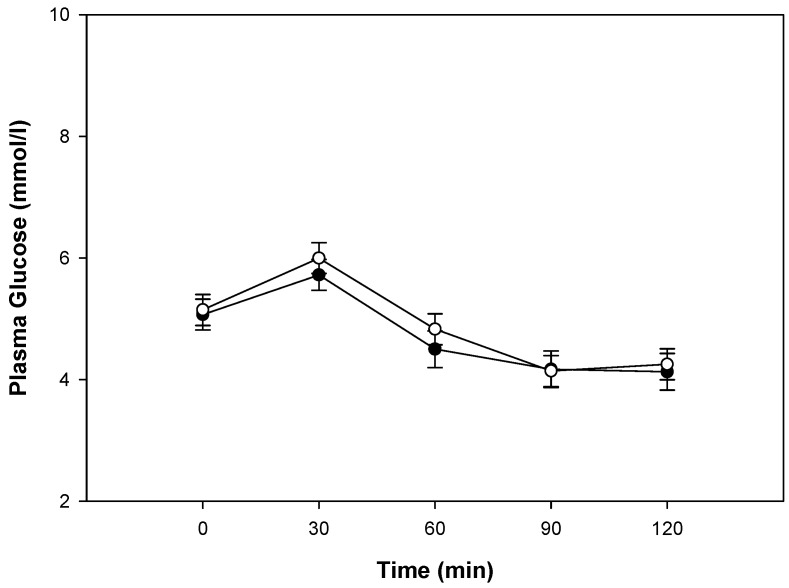
Plasma glucose and serum insulin concentrations after consumption of the blueberries (●) or placebo gel (○) breakfast meals. Data are presented as arithmetic means ± SE, *n* = 17. All times points were significantly different from baseline, *p* < 0.05. For glucose and insulin there were no significant differences between responses from the blueberry or placebo breakfast meals.

**Figure 3 nutrients-11-00202-f003:**
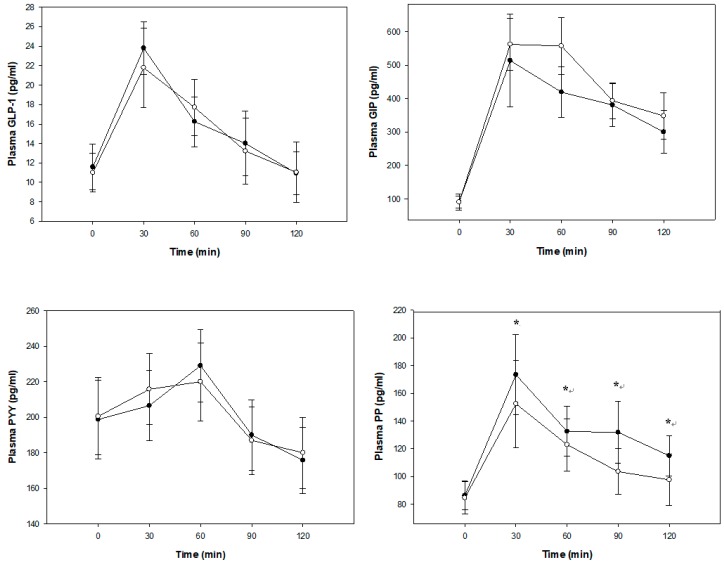
Gastrointestinal hormone concentrations after consumption of the blueberries (●) or placebo gel (○) breakfast meals. Data are presented as arithmetic means ± SE, *n* = 17. All times points were significantly different from baseline, *p* < 0.05. For glucagon-like peptide-1 (GLP-1), glucose-dependent insulinotropic peptide (GIP), and peptide YY (PYY) there were no significant differences between responses from the blueberry or placebo breakfast meals; pancreatic polypeptide (PP) responses were significantly different after consumption of the blueberry breakfast meal than the placebo breakfast meal, *p* < 0.05*.

**Figure 4 nutrients-11-00202-f004:**
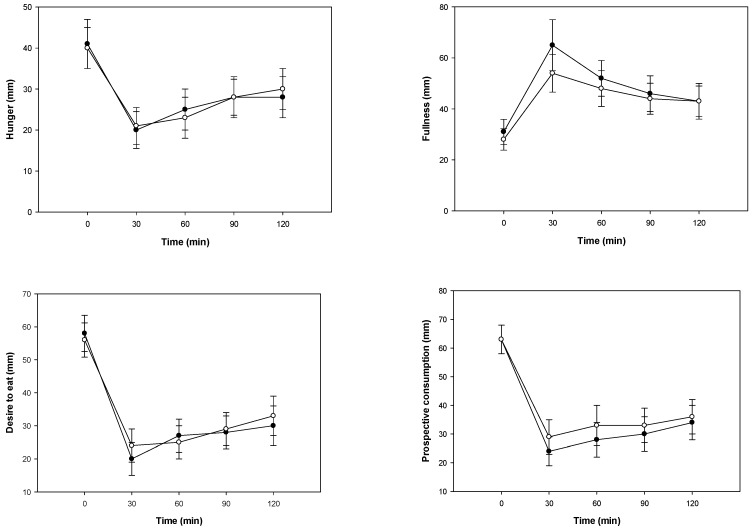
Perceived appetite response after the consumption of blueberries (●) or placebo gel (○) breakfast meals. Data are presented as arithmetic means ± SE, *n* = 17. All times points were significantly different from baseline, *p* < 0.05. For hunger, fullness, desire to eat and prospective consumption, there were no significant differences between responses from the blueberry or placebo breakfast meals.

**Table 1 nutrients-11-00202-t001:** Nutrient composition of whole blueberries and placebo gel treatment ^1^.

	Treatment Beverage	
	Whole Blueberries	Placebo Gel
Energy, kcal	80	78
Water, g	120	117
Fat, g	0.24	0
Protein, g	0	0
Carbohydrate, g	18	18
Sugars, g	10.3	11.1
Sucrose, g	0	0
Glucose, g	4.8	5.3
Fructose, g	5.5	5.8
Fiber, g	6.2	6.2
Phenolics, g	720	0
Anthocyanins, mg	401	0

^1^ Values represent content per 140 g of whole blueberries and placebo gel.

**Table 2 nutrients-11-00202-t002:** Characteristics of study participants at baseline ^1^.

	Total ^2^	Range
Age (years)	47 ± 15	21–63
Height (cm)	165.3 ± 12.3	143.0–188.0
Weight (kg)	64.3 ± 13.1	48.0–96.2
Body mass index (kg/m^2^)	23.4 ± 3.0	19.5–30.2
Body fat (%)	23.2 ± 7.0	13.1–34.8
Systolic blood pressure (mm Hg)	115.9 ± 6.3	102–124
Diastolic blood pressure (mm Hg)	75.1 ± 7.3	55–86
Glucose (mmol/L)	4.9 ± 0.3	4.1–5.4
Insulin (pmol/L)	38.5 ± 14.2	14.0–66.0
Subjective physical activity level (%)		
Mild exercise	12	
Occasional vigorous exercise	29	
Regular vigorous exercise	59	
Ethnicity (%)		
White, not Hispanic	100	

^1^ All values are means ± SD, ^2^
*n* = 17; *n* =13 female, *n* = 4 male.

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
