# Peer review of "Postprandial Effects of Blueberry (Vaccinium angustifolium) Consumption on Glucose Metabolism, Gastrointestinal Hormone Response, and Perceived Appetite in Healthy Adults: A Randomized, Placebo-Controlled Crossover Trial"

_nutrients, 2019, doi:10.3390/nu11010202_

Reviewer 1 Report

Study is a really nice clear crossover design- congratulations, I hope it's followed at some point by a similar design with a larger group of subjects- to see if the trend of insulin depression persists.

Abstract

Line 36- change sentence to read: 'Polypetide concentrations were statistically significantly higher during' etc etc.

Line 40: there is an extra unnecessary dot

Line 47: currently: 'genetics, obesity overeating' suggest you add 'and low levels of physical activity'

Line 89 change 'are losing' to 'having lost' 

Line 259- start the x axis at 2 mmol (i.e. no need for he lower values)

Line 391: add longer human clinical trials with a larger N number are necessary

Author Response

Authors responses to reviewers:

Reviewer 1 (Comments to Authors)

Study is a really nice clear crossover design- congratulations, I hope it's followed at some point by a similar design with a larger group of subjects- to see if the trend of insulin depression persists.

Response to reviewer: Thank you for your comments.

Abstract

Line 36- change sentence to read: 'Polypetide concentrations were statistically significantly higher during' etc etc.

Response to reviewer: Thank you for your comment. We have added the information. Please see line 36.

Line 40: there is an extra unnecessary dot

Response to reviewer: We have removed the dot.

Line 47: currently: 'genetics, obesity overeating' suggest you add 'and low levels of physical activity'

Response to reviewer: Thank you for your comment. We have added the information. Please see lines 47-48.

Line 89 change 'are losing' to 'having lost' 

Response to reviewer: We have added the information. Please see line 96.

Line 259- start the x axis at 2 mmol (i.e. no need for he lower values)

Response to reviewer: Thank you for your comment. We have changed the figure. Please see Figure 2 Glucose.

Line 391: add longer human clinical trials with a larger N number are necessary

Response to reviewer: Thank you for your comment. We have added the information. Please see line 408.

Reviewer 2 Report

Nutrients (ISSN 2072-6643): Postprandial effects of blueberry (Vaccinium angustifolium) consumption on glucose metabolism, gastrointestinal hormone response and perceived appetite in healthy adults: a randomized, placebo-controlled crossover trial

Overview: Stote and colleagues studied the effect of blueberry added to a carbohydrate based meal compared with a calorie matched control. The major findings were that adding blueberries increased pancreatic polypeptide and had slight reductions in early phase insulin with no affect on glucose or perceived appetite. The paper is well written but lacks clinical utility at this time given healthy people were studied.

Abstract

p-values should be provided.

Introduction

Ln 42-43. Suggest focus on type 2 diabetes only given the majority of the paper appears to focus on etiology aligned with development of diabetes vs. type 1 diabetes, which relates more to genetics/virus.

A transition statement identifying the gap in the literature ought to be provide, perhaps around ln 70-71. Written as is it’s hard to know how this study adds to the literature.

A hypothesis should be provided.

Methods

A rationale for studying healthy people ought to be provided given an aspect of the study is to focus on glucose metabolism. Individuals with type 2 diabetes would have been of great interest to understand if blueberries aid reduction in glycemia post-meal for instance and how this compares directly with healthy individuals.

Insulin sensitivity and pancreatic insulin secretion are mentioned often in the paper. The authors should calculate these outcomes to determine if the treatment had an effect.

iAUC is calculated. Have the authors considered calculation of early vs. total phase iAUC for substrates and hormones? It appears that some outcomes, e.g. insulin or GIP are lower within the first hour. This could suggest differences in release of insulin vs. synthesis of new insulin post-meal.

C-peptide ought to be determined in order to make comment on insulin secretion (e.g. Glucose stimulated insulin secretion and disposition index) as suggested. Insulin is cleared by the liver up to about 80%. As such, it’s hard to know if changes in insulin are due to sensitivity or secretion without C-peptides (hepatic extraction could be determined too). 

GLP-1 appears to go up at 30 min. Have the authors calculated delta scores to determine if the change in GLP-1 relative to baseline is greater post-blue berry vs. placebo?

How was body fat determined? Should be included in the methods.

How active were these participants? Inclusion of data in Table 1 would be helpful.

When was the last exercise bout performed? This is important as exercise performed the night before or morning of could influence measures.

Why was ghrelin not measured given it is the only gut derived appetite stimulating hormone?

Results

Table 1 would be nice to include fasting insulin. Reason is based on Figure 2, it seems fasting insulin on average is around 40 uU/ml. This is quite high and suggests people are insulin resistance. In fact, the rise in insulin over 200 uU/ml at 30 min is very high given the glucose concentrations. This suggests the patients are not healthy as described but rather insulin resistant. Are the units correct or should they be pM?

Discussion

Body fat as a covariate increased certain appetite scores. Is it possible though that individuals with excess fat were more insulin resistant, and it’s really the insulin resistance, not the body fat per se that altered these values? Statistics based on insulin sensitivity calculations should be performed and discussed to mechanistically help understand how excess fat impacts appetite perceptions better.  Moreover, discussion on the relevance of this is warranted given perception is impacted but not hormones. How does excess body fat impact perceptions of appetite that are unique from the hormones?

Author Response

Reviewer 2 (Comments to Authors)

Stote and colleagues studied the effect of blueberry added to a carbohydrate-based meal compared with a calorie matched control. The major findings were that adding blueberries increased pancreatic polypeptide and had slight reductions in early phase insulin with no affect on glucose or perceived appetite. The paper is well written but lacks clinical utility at this time given healthy people were studied.

Response to reviewer: Thank you for your comments.

Abstract

p-values should be provided.

Response to reviewer: The p-value was added to the abstract.

Introduction

Ln 42-43. Suggest focus on type 2 diabetes only given the majority of the paper appears to focus on etiology aligned with development of diabetes vs. type 1 diabetes, which relates more to genetics/virus.

Response to reviewer: We have updated the manuscript to focus on type 2 diabetes. Please see lines 42-43.

A transition statement identifying the gap in the literature ought to be provide, perhaps around ln 70-71. Written as is it’s hard to know how this study adds to the literature.

Response to reviewer: We have updated the manuscript. Please see lines 71-75.

A hypothesis should be provided.

Response to reviewer: We have updated the manuscript to include a hypothesis. Please see lines 79-80.

Methods

A rationale for studying healthy people ought to be provided given an aspect of the study is to focus on glucose metabolism. Individuals with type 2 diabetes would have been of great interest to understand if blueberries aid reduction in glycemia post-meal for instance and how this compares directly with healthy individuals.

Response to reviewer: Thank you for the thoughtful comment. We agree that studying individuals with type 2 diabetes would be of great interest. We have updated the manuscript to include a rationale for studying healthy people. Please see lines 98-100.

Insulin sensitivity and pancreatic insulin secretion are mentioned often in the paper. The authors should calculate these outcomes to determine if the treatment had an effect.

Response to reviewer: We calculated the homeostasis model assessment of insulin resistance (HOMA-IR) in the study participants. We found the baseline mean and SD of the group to be 1.4 ± 0.4, which does not indicate insulin resistance in our study participants. In addition, HOMA-IR was not significantly affected by the blueberry treatments (P = 0.3926). BMI, gender, weight at enrollment, height, age, and percent body fat were not significant covariates for the HOMA-IR results.

References:

Matthews, D.R., et al., Homeostasis model assessment: insulin resistance and beta-cell function from fasting plasma glucose and insulin concentrations in man. Diabetologia, 1985. 28(7): p.412-9. 

Wallace, T.M. et al., Use and abuse of HOMA modeling. Diabetes Care, 2004. 28(6): p. 1487-95.

iAUC is calculated. Have the authors considered calculation of early vs. total phase iAUC for substrates and hormones? It appears that some outcomes, e.g. insulin or GIP are lower within the first hour. This could suggest differences in release of insulin vs. synthesis of new insulin post-meal.

Response to reviewer: We calculated the early vs. total phase iAUC for glucose, insulin and the gut hormones. No treatment effects were seen for calculating iAUC’s from 0-60 minutes for glucose, insulin and the gut hormones. BMI, gender, weight at enrollment, height, age, and percent body fat were not significant covariates. We did not include this information in the manuscript.

C-peptide ought to be determined in order to make comment on insulin secretion (e.g. Glucose stimulated insulin secretion and disposition index) as suggested. Insulin is cleared by the liver up to about 80%. As such, it’s hard to know if changes in insulin are due to sensitivity or secretion without C-peptides (hepatic extraction could be determined too).

Response to reviewer: Thank you for the insightful comment. We did not evaluate C-peptide concentrations.

GLP-1 appears to go up at 30 min. Have the authors calculated delta scores to determine if the change in GLP-1 relative to baseline is greater post-blue berry vs. placebo?

Response to reviewer: Yes, we calculated the delta scores (though our study biostatistician does not like the change from baseline because the magnitude of change can be dependent on the starting value; the biostatistician likes the baseline covariate approach better) to determine if the change in GLP-1 relative to baseline is greater post blueberry vs. placebo. There was no effect of blueberry treatment (P = 0.7085).

How was body fat determined? Should be included in the methods.

Response to reviewer: We added the information to the methods section. Please see line 148.

How active were these participants? Inclusion of data in Table 1 would be helpful.

Response to reviewer: We collected subjective exercise and activity level of study participants; 59% reported regular vigorous exercise (i.e., work or recreation 4 or more times/week for 30 minutes); 29% reported occasional vigorous exercise (i.e., work or recreation, < 4 times/week for 30 minutes); and 12% reported mild exercise (i.e., climbing stairs, walking 3 blocks, golf). We added the information to Table 2.

When was the last exercise bout performed? This is important as exercise performed the night before or morning of could influence measures.

Response to reviewer: We reviewed the study participants daily questionnaires; we found the last bout of exercise to be the day prior before 5:00 PM.

Why was ghrelin not measured given it is the only gut derived appetite stimulating hormone?

Response to reviewer: Thank you for the comment. We did not include ghrelin; we were interested in evaluating the other gut hormones. 
Results

Table 1 would be nice to include fasting insulin. Reason is based on Figure 2, it seems fasting insulin on average is around 40 uU/ml. This is quite high and suggests people are insulin resistance. In fact, the rise in insulin over 200 uU/ml at 30 min is very high given the glucose concentrations. This suggests the patients are not healthy as described but rather insulin resistant. Are the units correct or should they be pM?

Response to reviewer: The units of measurement for insulin concentrations are not correct. The measurement should be in pmol/l. We are sorry for the confusion. We have changed this in the manuscript, please see Figure 2 Insulin. In addition, we have added fasting baseline insulin to Table 2.

Discussion

Body fat as a covariate increased certain appetite scores. Is it possible though that individuals with excess fat were more insulin resistant, and it’s really the insulin resistance, not the body fat per se that altered these values? Statistics based on insulin sensitivity calculations should be performed and discussed to mechanistically help understand how excess fat impacts appetite perceptions better.  Moreover, discussion on the relevance of this is warranted given perception is impacted but not hormones. How does excess body fat impact perceptions of appetite that are unique from the hormones?

Response to reviewer: Thank you for the excellent comment. We calculated baseline HOMA-IR

for the study participants. No study participants were identified as having insulin resistance(having a HOMA-IR > 2.6). In addition, HOMA-IR was not a signficiant covariate for VAS or gut hormones. We agree that more research is needed to determine how appetite is perceived by individuals with insulin resistance.  

Reference: 

Matthews, D.R., et al., Homeostasis model assessment: insulin resistance and beta-cell function from fasting plasma glucose and insulin concentrations in man. Diabetologia, 1985. 28(7): p.412-9.

Reviewer 3 Report

As requested, I reviewed the manuscript (ID nutrients-419451) "Postprandial effects of blueberry (Vaccinium angustifolium) consumption on glucose metabolism, gastrointestinal hormone response and perceived appetite in healthy adults: a randomized, placebo-controlled crossover trial", by K Stote, A Corkum, M Sweeney-Nixon, N Shakerley, T Kean, K Gottschall-Pass.

The manuscript describes the effects of the bioactive content of blueberries on some key metabolic processes related to type 2 diabetes. The study is based on the addition of blueberries to a higher-carbohydrate breakfast meal and, consequently, on the analysis of postprandial glucose metabolism, gastrointestinal hormone response and perceived appetite in human healthy adults. Results show absence of variations of several gastrointestinal hormones, e.g. GLP-1, GIP, PYY, and describe parameters of the perceived appetite. Interestingly, variations of pancreatic polypeptide (PP) concentrations were shown to be occurring due to consumption of the blueberry breakfast meal.

My comment is in favor of publication. More in detail:

1)    Even though it's short, the Introduction sounds comprehensible and there’s no fragmented information. Also, it gives the necessary keywords to enter the investigated field. Nevertheless, it can be noticed that some more information on the peptides investigated in the research might render the manuscript even more appreciable.

2)    Methods are very well detailed thus leaving no doubts and represent a guideline for similar approaches; Results are clear; Discussion is convincing and exhaustive.

On this basis, the authors demonstrated appreciable scientific mastery of their research.

I add below two (very) minor suggestions for reaching a better final shape.

Minor comments:

a)    The Introduction is quite concise thus authors could improve the information content by adding a short overview of all the investigated peptides. This might help the readers to better appreciate the results which show how PP peptide differs from the others.

b)    In the Discussion/Conclusions, the importance of longer human clinical trials could be better shaped/described besides the obvious necessity. For instance, mid-to-long-term detection of glucose, insulin, (all) peptides and appetite parameters might disclose interesting aspects of specific oscillation around the setpoints that could reveal actual effects on metabolic shifts induced by this category of compounds.

Thank you very much for your attention to my opinion.

Author Response

Reviewer 3 (Comments to Authors)

As requested, I reviewed the manuscript (ID nutrients-419451) "Postprandial effects of blueberry (Vaccinium angustifolium) consumption on glucose metabolism, gastrointestinal hormone response and perceived appetite in healthy adults: a randomized, placebo-controlled crossover trial", by K Stote, A Corkum, M Sweeney-Nixon, N Shakerley, T Kean, K Gottschall-Pass.

The manuscript describes the effects of the bioactive content of blueberries on some key metabolic processes related to type 2 diabetes. The study is based on the addition of blueberries to a higher-carbohydrate breakfast meal and, consequently, on the analysis of postprandial glucose metabolism, gastrointestinal hormone response and perceived appetite in human healthy adults. Results show absence of variations of several gastrointestinal hormones, e.g. GLP-1, GIP, PYY, and describe parameters of the perceived appetite. Interestingly, variations of pancreatic polypeptide (PP) concentrations were shown to be occurring due to consumption of the blueberry breakfast meal.

My comment is in favor of publication. More in detail:

1)    Even though it's short, the Introduction sounds comprehensible and there’s no fragmented information. Also, it gives the necessary keywords to enter the investigated field. Nevertheless, it can be noticed that some more information on the peptides investigated in the research might render the manuscript even more appreciable.

2)    Methods are very well detailed thus leaving no doubts and represent a guideline for similar approaches; Results are clear; Discussion is convincing and exhaustive.

On this basis, the authors demonstrated appreciable scientific mastery of their research.

Response to reviewer: Thank you very much for your comments.

I add below two (very) minor suggestions for reaching a better final shape.

Minor comments:

a) The Introduction is quite concise thus authors could improve the information content by adding a short overview of all the investigated peptides. This might help the readers to better appreciate the results which show how PP peptide differs from the others.

Response to reviewer: Thank you very much for your comment. We have added additional information about the gastrointestinal hormones in the introduction section. Please see lines 69-70.

b)    In the Discussion/Conclusions, the importance of longer human clinical trials could be better shaped/described besides the obvious necessity. For instance, mid-to-long-term detection of glucose, insulin, (all) peptides and appetite parameters might disclose interesting aspects of specific oscillation around the setpoints that could reveal actual effects on metabolic shifts induced by this category of compounds.

Response to reviewer: Thank you very much for your comment. We have added more information in the Discussion/Conclusion sections. Please see lines 352-354, 383-384 and line 414.

Thank you very much for your attention to my opinion.

Round  2

Reviewer 2 Report

Congrats on a nice paper.